# Trouble for Horses in Paradise: Toxicity and Fatality Resulting from the Consumption of *Indigofera spicata* (Fabaceae) on Oahu Island

**DOI:** 10.3390/vetsci9060271

**Published:** 2022-06-04

**Authors:** Mohsen Mohamad Ramadan, Devon Dailey

**Affiliations:** 1Plant Pest Control Branch, Division of Plant Industry, Hawaii Department of Agriculture, Honolulu, HI 96814, USA; 2Hawaii Polo Club and Hawaii Polo Trail Rides, Waialua, HI 96791, USA; hawaiipolo@gmail.com

**Keywords:** *Indigofera spicata*, Oahu island, horse toxicity, invasive weed

## Abstract

This is the first account of fatal toxicity in horses resulting from grazing on the pasture plant creeping indigo, *Indigofera spicata*, on the island of Oahu, in the Hawaiian Islands. A survey in the town of Waialua on the north shore of Oahu island indicated that creeping indigo is common and abundant on grazing lands during the rainy season and requires intensive chemical and physical control measures. Four pastures were surveyed where ranchers reported mortality of more than 17 horses since 2020. We document these incidents to alert state animal and livestock support officials, groups monitoring invasive species, and horse owners regarding the problem of this noxious weed and to support breeders with information to confront its invasiveness. Herbicide treatment is not economically feasible, and breeders opted to physically uproot the plants from the paddocks and restrain horses to clear pastures as they were eliminating the plants. We urge state officials for a long-term control strategy to reduce the problems associated with this weed.

## 1. Introduction

### 1.1. Background

Creeping indigo, *Indigofera spicata* Forsskal (Fabaceae) is a perennial herb native to tropical Africa, Madagascar, and the Mascarene islands, with widespread prevalence in grazing pastures throughout tropical Asia and Australia [1]. In Hawaii, it was introduced prior to 1929 as a pasture legume, and soon appeared to be poisonous to livestock. Creeping indigo is an aggressive, drought-tolerant, relatively palatable prostrate legume. Currently, it is adapted to low elevation, dry, and disturbed areas on all major Hawaiian Islands [2,3].

Creeping indigo has become invasive in many regions where it was introduced as a forage or cover crop, such as in Puerto Rico and southern Florida [4]. Because it is highly appealing to livestock, horses may be poisoned after eating large amounts of the legume. It carries a high level of toxic alkaloids that can cause miscarriage in pregnant cattle, among other effects [5,6].

### 1.2. Plant Identity, Origin, and Distribution

Plants of the tribe *Indigofereae* (Fabaceae) are often very attractive, pink- to red-flowered shrubs and herbs. The largest genus is *Indigofera* (≈700 species) which is pantropical, with two-thirds of the species restricted to Africa, the rest in the Sino-Himalayan subregion, Indonesia, and Brazil. Because of the complex chemistry of this genus, it contains many species that are poisonous or important in agroindustry [2]. For instance, the original source of blue indigo dye is *Indigofera tinctoria* L., a shrub cultivated in tropical and temperate Asia and Africa [1].

*I. spicata*, also known as lawn indigo or trailing indigo, is an herbaceous plant with prostrating, slightly flattened and sparsely hairy stems that branch out from a central root. The leaves are usually light to dark green in color, with 5–11 oblong leaflets per stem arranged in an alternating pattern (0.5–3.0 cm long) with sparsely hairy upper surfaces and dense hairs on the undersides. The flowers are small, red or pink, and pea-shaped (0.5 cm long), arranged in elongated clusters up to 10 cm long, and produce abundant narrow, cylindrical, straight, sharp-tipped seed pods (1.5–3.5 cm long) (Figure 1 and Figure 2). This species reproduces mainly by seeds dispersed by lawn mowers, contaminated soil, water, and mud attached to animals and vehicles. It is a very common weed of lawns, gardens, roadsides, disturbed sites, and waste areas [2].

*I. spicata* was valued as a cover crop to manage soil quality in coffee lands in Africa and was introduced into India, Java, Malaya, and the Philippines as an ornamental ground cover crop in tea, rubber, oil palm, and sisal plantations [1]. It has also been introduced to Australia, several Pacific islands, central America, and the Caribbean islands [1,2,3].

### 1.3. Symptoms, Toxicity, and Toxic Components

Creeping indigo grows very close to the ground, which makes it difficult to find it within a pasture (Figure 1A). Beside growing from its seeds, it grows back from its long taproots, which makes it hard to destroy it with a single application of chemicals (Figure 1B) [1,2].

Two putative toxins found in creeping indigo are 3-nitropropionic Acid (3-NPA) and indospicine. The highly toxic 3-NPA is a potent and irreversible inhibitor of mitochondrial succinate dehydrogenase, a key enzyme in transforming glucose and oxygen into useable energy, accounting for the early and prominent neurologic signs of toxicity. Because 3-NPA is metabolized quickly, it is unlikely to be found when testing the serum of affected animals. It is associated with motor disorders in livestock and humans when ingested [7,8].

Indospicine is a non-protein amino acid toxic to the liver because of its antagonism to arginine, an essential α-amino acid that is important in the biosynthesis of proteins. One of its principal toxic actions is the inhibition of nitric oxide synthase, an action associated with the development of ulceration of mucous membranes [9], Figure 3B. Although horses are relatively resistant to the liver-damaging effects of indospicine, this toxin accumulates in the tissues of horses dying from the disease. Indospicine accounts for ≤0.5% of the dry matter of creeping indigo and can be detected in the serum of affected animals [10]. It is highly toxic to livestock in small doses, causing loss of vitality, liver degeneration, and abortion in cattle and goats. It is particularly dangerous to horses, which relish plants containing it [10,11]. Horses need to eat around 4.5 kg of creeping indigo daily for about two weeks to develop signs of toxicity. Symptoms vary but can include runny squinting eyes, sleepiness, involuntary rapid and repetitive movements of the eyes, abnormal gait, and mild colic-type signs. Special tests on postmortem tissues can identify the toxin [12,13,14].

Ossedryver et al., 2013 [15], reported that horses that continuously grazed a pasture containing about 24% *I. spicata* for 4–6 weeks developed weakening, lack of muscle coordination, depression, swallowing disorder, excessive flow of saliva, and bad breath. One horse in the study recovered with supportive treatment, but two other horses were hopelessly sick. Their livers showed lymphocytic infiltrations and hydropic degeneration of hepatocytes, with mild to moderate liver necrosis. Indospicine was detected in the serum, heart, and liver of euthanized horses. 3-NPA and indospicine toxins were believed to have caused neurological syndrome and the formation of indospicine residues in the tissues of animals grazing paddocks infested with *I. spicata* in Australia [15].

### 1.4. Signs of Creeping Indigo Toxicity

Early neurological signs include a change in performance in affected horses, with individuals becoming less energetic than usual and showing degrees of sleepiness ranging from mild exhaustion to loss of consciousness as the condition progresses over days to weeks. Horses also show episodes of standing sleep-like activity, head-pressing into corners, or compulsive walking around the inside of a stall or paddock. Some affected horses have been seen with their heads tilted to one side and their necks and bodies twisted in the same direction, indicating involvement of the balance centers of the brain. Most horses that continue to consume the plant eventually become unconscious or develop convulsions and mild liver pathology [13,14].

Non-neurologic signs may include weight loss, high heart and respiratory rates, labored breathing, high body temperature, foaming from the mouth, dehydration, pale mucous membranes, watery discharge from the eyes and squinting, light sensitivity, corneal ulceration and severe ulceration of the tongue and gums, and acute diarrhea [16], Table 1, Figure 3A,B.

### 1.5. Current Records of Toxicity and Fatality on Oahu Island

Table 2 presents horse toxicity from creeping indigo at the Hawaii Polo Club on the North Shore of Oahu (21°34′42.42″ N, 158°10′18.41″ W, elevation 0.9–4.2 m) paddocks to alert ranchers and state animal health officials of this problem and to enable instruction on methods for reducing weed population. Veterinary records showed high Gamma Glutamyl Transferase enzyme (GGT) tallies for several horses were improved after isolating the horses to clean paddocks (Table 2). A rancher in Waialua, Dillingham Ranch (21°34′20.19″ N, 158°10′27.45″ W, elevation 6.0 m), reported horse deaths and illnesses on the North Shore, likely from creeping indigo feeding. He also testified regarding three horses currently experiencing toxicity and elevated liver function tests, and the loss of one horse from liver failure in 2017 (Taylor Mower, Dillingham Ranch Manager, personal communication). Four pastures were surveyed where ranchers reported added mortality of more than 17 horses since 2020.

### 1.6. Chemical Control

Although there is no commercial indigo-selective herbicide recommended for creeping indigo to kill leaves and roots, two herbicides containing aminopyralid are used, with necessary retreatment the following year. One of these is Milestone (DOW AgroSciences) at 5 fluid ounces per acre. Hawaii Polo Club on Oahu uses GrazonNext HL DOW herbicides for control of broad-leaf weeds on rangelands at 24 fluid ounces (= 0.7 L) per acre [17]. Plants die if sprayed with Grazon or other similar products, but pulling up the dead plants is necessary since the seeds are still viable (Figure 2). The dead plants must be removed and disposed [18].

### 1.7. Treatment and Prevention

There is no effective treatment for advanced toxicity by creeping indigo, only the management of symptoms is possible. Horses that are quickly removed from infested sites often recover completely, but gait abnormalities sometimes persist. Because of their relative resistance to indospicine, horses lack substantial liver lesions. Combined with the neurological disease that results from 3-NPA poisoning, this suggests that L-arginine alone would have little benefit in the treatment of the neurological symptoms [19]. Thiamine was also cited in the literature as a current treatment for nitro toxicity in livestock, but research showed this treatment is ineffective [19]. Therefore, the management of affected horses should emphasize their removal from the source of toxicity to prevent any injuries. It was suggested that livestock poisoning by *I. spicata* can be prevented by reducing the weed population to <20% of the total forage available, but recent evidence does not support this view. The confinement of horses to clean paddocks where creeping indigo is removed by physical means or herbicide application is the best control measure [13,20].

## 2. Discussion

Several species of the genus *Indigofera* (e.g., *I. hirsute* L., *I. pilosa* Poir., *I. schimperi* Jaub. & Spach, and *I. subulate* Vahl ex Poir.) have shown great potential as high-protein grazing forage, green manure, or cover crops for livestock in Africa and Asia [21,22]. Some *Indigofera* species are known to contain the pigment indigo used for commercial dye production. However, reservations regarding the toxicity of *I. spicata* have restricted its establishment as a feed supplement [21]. It was introduced to Hawaii as a pasture legume to supplement the diets of grazing animals, but its toxicity when consumed in large quantities was unknown before importation.

The introduction of the plant and seeds to Hawaii at that time was lawful, with no detailed risk analysis by the Animal and Plant Health Inspection Service (APHIS). Currently, creeping indigo is thriving in low-elevation, dry, disturbed areas [2]. It was initially introduced under the name *I. hendecaphylla* Jacq. [23], a common species, distributed throughout the Old-World tropics and subtropics to the Pacific Islands, whereas *I. spicata* is confined to Africa, Yemen, Madagascar, and the Mascarenes [1]. Previous reports on the toxicity of creeping indigo as a cattle fodder referred to *I. hendecaphylla* [23,24]. *I. spicata* has also been shown to be hepato-toxic when grazed by cattle [5]. The free, non-protein amino acid indospicine was detected in the seed and leaf material of this plant. Besides *I. spicata,* toxicity was also reported for the African species *I. hirsute* L. and *I. linifolia* (L.f.) Retz. [21,22].

The enzyme gamma glutamyl-transferase (GGT) is generally found in liver cell membranes and biliary epithelial cells. Increased blood levels of GGT indicate that something is damaging the liver, though they do not identify the specific problem (Table 2). Increased levels of GGT can take several weeks before returning to normal. Higher-than-normal test results could be a sign of liver disease such as hepatitis, cirrhosis, tumors, or pancreatic cancer [25,26].

The Hawaii Polo Club had multiple cases of suspected alkaloid toxicity in horses. Field inspection of four pastures revealed that the original suspect was *I. spicata* mixed with the nutritious kikuyu grass, *Cenchrus clandestinus* Hochst. ex Chiov. (Poaceae). Other invasive weeds were common in paddocks but disregarded as the culprit of horse toxicity [e.g., Ivy gourd, *Coccinia grandis* (L.) Voigt (Cucurbitaceae); wild tobacco bush, *Nicotiana glauca* Graham (Solanaceae); Golden crown-beard, *Verbesina encelioides* (Cav.) Benth. & Hook. f. ex Gray (Asteraceae); Coat buttons, *Tridax procumbens* (L.) (Asteraceae), M. Ramadan unpublished data]. Chronic pyrrolizidine alkaloidosis as in fireweed, *Senecio madagascariensis* Poir. (Asteraceae), causes bile duct hyperplasia and biliary disease and therefore typically results in raised serum GGT levels. Fireweed was not found during the survey on Oahu ranches, it is only established in pastures on Hawaii and Maui islands [27].

## 3. Conclusions

This synopsis of the invasive plant *I. spicata* on Oahu island indicates a recent increase in toxicity and fatality in horses on four ranches. The plant is widespread on pasture lands of windward Oahu during the rainy season. Chemical control with broad-spectrum herbicides appears ineffective, and manually pulling the plants is not sufficient for managing this weed. In general, a single herbicide application will not suppress creeping indigo permanently; research on chemical control is essential for new selective herbicides that kill leaves and taproots and do not injure useful pastures.

We urge state officials and Regulatory Enforcement and Animal Care (REAC) program USDA–APHIS to fund research on long-term solutions to decrease the plant population. Creeping indigo is very palatable to horses, and it is important for horse owners to recognize the plant and know the signs of toxicity and ways to prevent it from being consumed. Statewide surveys and outreach programs are needed to accurately determine the infestation levels of *I. spicata* and unreported livestock toxicity. This weed should be elevated to the top list of noxious weeds targeted for management in Hawaii. The Hawaii Department of Agriculture may participate in a subsidy program to assist ranchers in chemical control to mitigate this problem until biological control agents are found. A biological control endeavor in the native region is recommended for importation of host-specific natural enemies to reduce its invasiveness. For instance, the wild indigo weevils, *Trichapion rostrum* (Say), (Coleoptera: Brentidae), and *Tychius sordidus* LeConte, (Coleoptera: Curculionidae), are seed predators used in biological control and can adversely affect the reproduction of related herbaceous Fabaceae plants [28]. Several other natural enemies (i.e., insects, nematodes, and diseases) of *Indigofera* can be examined for specificity.

## Figures and Tables

**Figure 1 vetsci-09-00271-f001:**
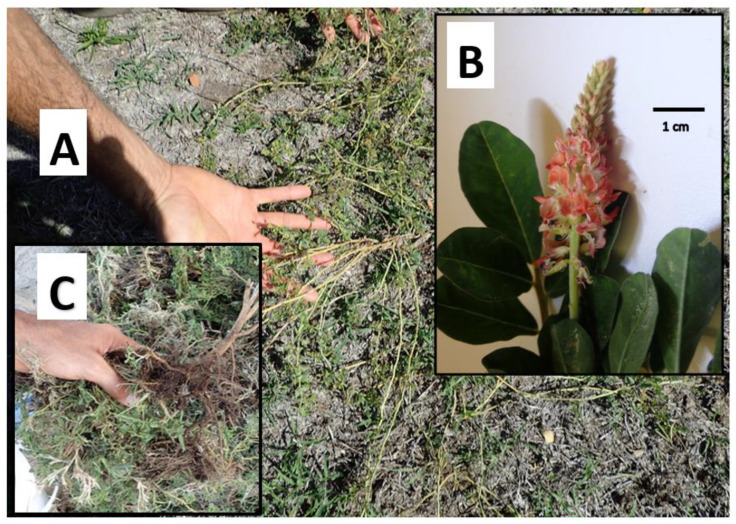
Creeping indigo, *Indigofera spicata*, filling pastures on Oahu island: (**A**) creeping indigo mature plant with stems creeping along very close to the ground, making it hard to notice among other foliage, (**B**) bunch showing pink flowers and the oblong, pinnate leaves typical of Fabaceae, (**C**) uprooted plant showing tough deep central roots that are difficult to kill with conventional herbicides and must be dug out. Creeping indigo also spreads by its long, hard-to-pull roots and seeds, making it difficult to fully kill it in one round.

**Figure 2 vetsci-09-00271-f002:**
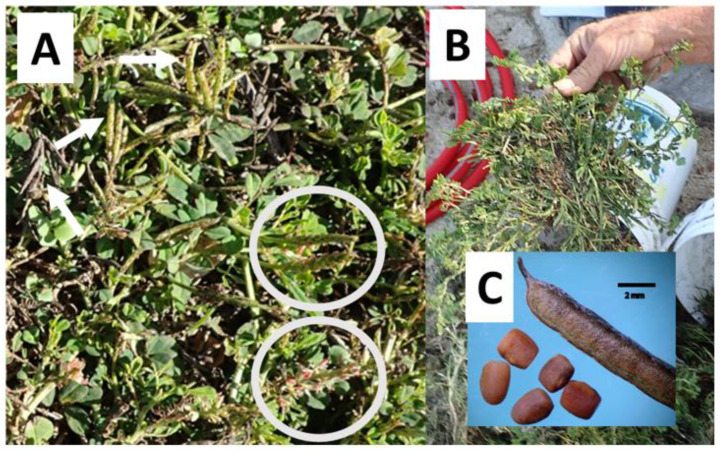
Creeping indigo on Oahu island, Waialua Polo Ranch: (**A**) the white circles enclose pink inflorescences arranged in elongated clusters (≈10 cm long) with oblong dark green leaves typical of Fabaceae, with 5–11 leaflets per stem arranged in an alternating pattern (0.5–2.5 cm long). The white arrows point to clusters of bright green immature and dry narrow straight cylindrical pods (≈2.5 cm long); (**B**) uprooted creeping plant; (**C**) typical hairy stiff dry pod with pointed hooked tip, contains 4–8 light brown quadrangular ≤2 mm-long seeds (bar = 2 mm).

**Figure 3 vetsci-09-00271-f003:**
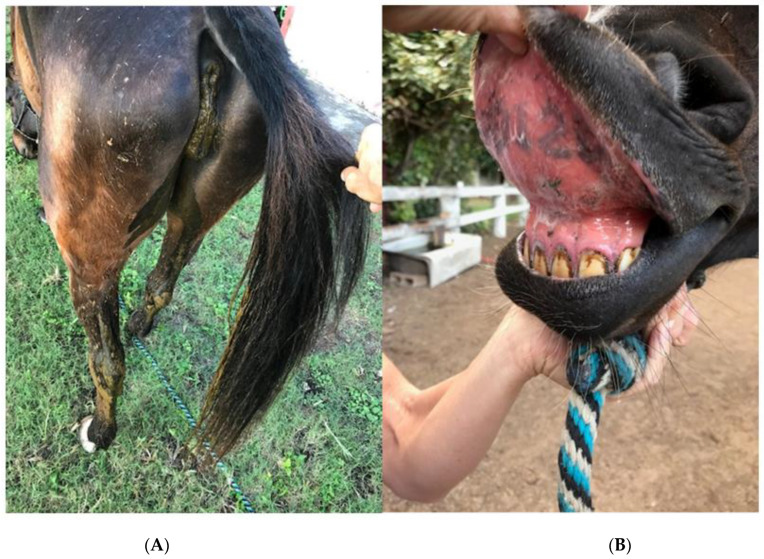
Symptoms of the late horse Serena: weight loss, lethargy, fasting, diarrhea (**A**); red gums with extensive ulcerations above the upper teeth and tongue (**B**) of the horse “Serena” before she died on 20 November 2020, on the north shore of Oahu island with symptoms of creeping indigo toxicity.

**Table 1 vetsci-09-00271-t001:** Veterinary transcript, owner’s notes, and treatment for the late horse Serena two days before death. Hawaii Polo Club and Hawaii Polo Trail Rides data. Source: unpublished diagnostic report, Dr. Manuel G. Himenes, ABVP Oahu Equine Veterinary Clinic.

Date and Hour	Symptoms	Treatment	Notes
18 November 2020 10:23 a.m.	Stopped eating last night	7 ounces of molasses (198 g)	
19 November 2020 12:22 a.m.	Breathing heavily, dehydrated Fever 105.3	2 ounces of molasses (56 g)	
12:49 a.m.	Pulse 92.0		
1:15 a.m.	Pulse 104.7	10 cc Banamine 4 g Bute	Equine ice boots on, pacing
1:36 a.m.	Pulse 104.9		
1:58 a.m.	Pulse 102.9		
2:30 a.m.	Pulse 105.0		
3:27 a.m.	Pulse 98.5		After hosing and icing for 15–20 min.

**Table 2 vetsci-09-00271-t002:** Veterinary serum chemistry reports and chronology of GGT levels of Hawaii Polo Club’s 20 horses after isolation to clean paddocks. Hawaii Polo Club and Hawaii Polo Trail Rides blood test data 2020–2021 (* Indicates horse dead). Source: unpublished diagnostic report, Dr. Manuel G. Himenes, ABVP Oahu Equine Veterinary Clinic.

Horse	Date (m/d/y)	^1^ GGT (IU/L) Enzyme Level
Brutus	1 November 2021	28
Haole Girl	1 November 2021	WNL
Truco	19 October 2021	56
“	18 September 2021	22
Molly	19 October 2021	116
“	13 March 2021	WNL
Mia	19 October 2021	WNL
Jackson	19 October 2021	23
“	29 June 2021	WNL
“	13 March 2021	WNL
John Wayne	15 October 2021	239
Rincon	24 August 2021	203
“	12 July 2021	259
“	8 June 2021	104
“	6 March 2021	238
Bingo	21 May 2021	121
Siracha	5 April 2021	25
“	9 November 2021	WNL
Hoagi	19 March 2021	WNL
Holly	13 March 2021	WNL
Cali	9 November 2021	WNL
“	13 March 2021	41
Blueberry	13 March 2021	WNL
Kupaa	5 January 2021	46
“	22 December 2020	56
“	27 December 2020	WNL
Pia	22 December 2020	WNL
Serena *	17 November 2020	131
“	7 November 2020	155
Tomahawk	9 November 2021	WNL
Apollo	9 November 2021	WNL
Anty Up	9 November 2021	WNL

^1^ GGT = Gamma Glutamyl Transferase enzyme, IU/L = international units per liter, WNL = within normal limits (“) = same as above.

## Data Availability

The data presented in this study are available on request from the corresponding author.

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
