# Peer review of "Trouble for Horses in Paradise: Toxicity and Fatality Resulting from the Consumption of Indigofera spicata (Fabaceae) on Oahu Island"

_vetsci, 2022, doi:10.3390/vetsci9060271_

Round 1
Reviewer 1 Report
It is evident that the authors have had native English speakers review and edit the original manuscript and it has improved greatly. There are still a few awkward sentences here and there but one more inspection should find and correct them. Furthermore, page 4, lines 113-116 are single spaced and bold, was this intentional?
The tables now are cohesive in appearance; however, the figures are problematic. They still are overlapping and obscuring each separate image for what they are, red arrows are still on the green image (preventing color blind individuals from observing them), the text font was changed but is still difficult to read and since these are perhaps enlarged, the images are blurred (e.g., figure 1 B and C). Also, page 2, line 74, lists a reference to Figure 3B for the lines 73-74, however, the image isn’t clearly showing a pathway or development of disease. There is no comparison to a normal mucous membrane or something to show how this is different. The figures are still unacceptable as is.
References are needed in several places: page 1, lines 34-35; page 2, lines 57-58 and 58-59; page 2, lines 63-64; page 2, lines 76-77; page 2, line 77-79; page 11, lines 173-175; page 11, lines 184-185; page 11, lines 210-213. Both tables contain data from a source, but the source does not have a reference listed in the reference section, this needs to be added. Also, page 2, line 70, refers to “motor disorders in livestock and humans” but only lists one reference which is concerning chicks. Please add further references to include humans.
In reexamining the manuscript, I find that a map is a necessary component. One overall of the islands indicating which of the Hawaiian Islands Oahu is, and specifically Oahu, where the four ranches were generally situated. This would then give a better indication on how well I. spicata is spread throughout a region.
The Conclusions end with statements that chemical control and manually pulling isn’t adequate. The authors recommend state officials find a long-term solution to the problem. This is perplexing since the authors are far more knowledgeable in the subject matter and could come up with theoretical solutions to present as possibilities here. It seems the notion of a biological control endeavor of a host-specific natural enemy is rather ambiguous and inconclusive. It would be better to postulate than simply leave the situation empty.
Author Response
Responses to reviewers
The authors would like to thank the anonymous reviewers for careful reviews reading the manuscript twice. I revised the manuscript using all his suggestions (in blue). Thanks to reviewer 1, below are my responses (in blue) to his comments (in black):
- It is evident that the authors have had native English speakers review and edit the original manuscript and it has improved greatly. There are still a few awkward sentences here and there but one more inspection should find and correct them. Furthermore, page 4, lines 113-116 are single spaced and bold, was this intentional?
I am sorry, cannot find this error that may be corrected through MDPI format.
- The tables now are cohesive in appearance; however, the figures are problematic. They still are overlapping and obscuring each separate image for what they are, red arrows are still on the green image (preventing color blind individuals from observing them),
Thank you. Separate images within the graphs are now framed, letters and arrows are now larger filled white for clarity.
- the text font was changed but is still difficult to read and since these are perhaps enlarged, the images are blurred (e.g., figure 1 B and C). Font letters are enlarged and framed in white. Figure 1 B and C changed.
- Also, page 2, line 74, lists a reference to Figure 3B for the lines 73-74, however, the image isn’t clearly showing a pathway or development of disease. There is no comparison to a normal mucous membrane or something to show how this is different. The figures are still unacceptable as is. To clear this, a new reference added contains photos of mucous membranes of normal and sick horses (The Brooke, 2013. Animal Health Mucous Membranes. The Working Equid Veterinary Manual; Whittet Books, Essex. 9 pp. https://www.thebrooke.org/sites/default/files/Animal%20Welfare/Welfare%20Interpretation%20Manual-%20Chapter%202%2C%20MucousMembranes.pdf accessed May 17 2022. ).
- References are needed in several places, page 1, lines 34-35;
[4] Morton, J. F., 1989. Creeping Indigo (Indigofera spicata Forsk.) (Fabaceae) - A hazard to herbivores in Florida. Econo. Bot., 43 (3): 314-327
https://doi.org/10.1007/BF02858731
- lines 57-58 and 58-59; page 2,
added reference [1,2,3]
- lines 63-64; page 2,
reference [1,2] added
- lines 76-77; page 2,
reference [8] added
- line 77-79; page 11,
reference [9] added
- lines 173-175; page 11,
reference [17] added.
- lines 184-185; page 11,
reference [18, 19] added
- lines 210-213.
Added personal observations (M. Ramadan unpublished data.)
- Both tables contain data from a source, but the source does not have a reference listed in the reference section, this needs to be added.
Source added to table titles: (Source: unpublished diagnostic report, Dr. Manuel G. Himenes, ABVP Oahu Equine Veterinary Clinic.)
- Also, page 2, line 70, refers to “motor disorders in livestock and humans” but only lists one reference which is concerning chicks. Please add further references to include humans.
New reference added: [27] Bendiksen Skogvold H, Yazdani M, Sandås EM, Østeby Vassli A, Kristensen E, Haarr D, Rootwelt H, Elgstøen KBP. A pioneer study on human 3-nitropropionic acid intoxication: Contributions from metabolomics. J Appl Toxicol. 2022 May;42(5):818-829. doi: 10.1002/jat.4259. Epub 2021 Nov 1. PMID: 34725838.
- In reexamining the manuscript, I find that a map is a necessary component. One overall of the islands indicating which of the Hawaiian Islands Oahu is, and specifically Oahu, where the four ranches were generally situated. This would then give a better indication on how well I. spicata is spread throughout a region.
I agree, I added a reference for distribution map of this weed over all major Hawaiian island has been reported in CABI.org https://www.cabi.org/isc/datasheet/79262. Therefore, I only added the GPS data for the specific locality surveyed in this report.
- The Conclusions end with statements that chemical control and manually pulling isn’t adequate. The authors recommend state officials find a long-term solution to the problem. This is perplexing since the authors are far more knowledgeable in the subject matter and could come up with theoretical solutions to present as possibilities here. It seems the notion of a biological control endeavor of a host-specific natural enemy is rather ambiguous and inconclusive. It would be better to postulate than simply leave the situation empty.
I agree and add few statements in conclusions regarding theoretical solutions and possibilities to mitigate the weed problems (in blue). Research fund is needed for new chemicals that kill the roots, which are difficult to kill with conventional herbicides. Also added examples of natural enemies currently used for biological control against related weeds.

Reviewer 2 Report
Ramadan et al. resubmitted the manuscript titled "Trouble for Horses in Paradise: Toxicity and fatality resulting from consumption of Indigofera spicata (Fabaceae) on Oahu island", which have addressed major comments during initial review and is now improved over previous version. This manuscript qualifies for publication, and can be accepted for publication in MDPI Veterinary Sciences.
Author Response
The authors would like to thank this reviewer for his time reading our manuscript twice.
Aloha and Mahalo
Round 2
Reviewer 1 Report
All updates and changes from review comments seem to have improved the manuscript. Minor spelling and grammar double check is the last edit recommended. No further comments needed.
Author Response
Many thanks for anonymous reviewer. I checked spelling and grammar using word editor.
Aloha and Mahalo,
Mohsen
This manuscript is a resubmission of an earlier submission. The following is a list of the peer review reports and author responses from that submission.
Round 1
Reviewer 1 Report
Ramadan and Daily submitted a manuscript titled "Trouble to Horses in Paradise: toxicity and fatal cases by Indigofera spicata (Fabaceae) on Oahu Island." for publication in MDPI Veterinary Sciences.
Though the scope of the journal allows this manuscript, the quality of writing has to be extensively improved. The language part also needs extensive improvement.
I suggest authors to seek professional help if required to improve the quality of writing. Few examples:
Line 2: Capitalize "Toxicity". Title needs rephrasing.
Line 26: remove capitalization of "Islands".
line 28: the sentence "Now naturalized in low elevations, dry and disturbed areas" has no meaning. Several sentences like this are either incomplete or unstructured.
Line 33: delete "as"
Line 33: "Because it is highly appetizing.." sentence needs rephrasing.
The manuscript has only two sections: Background and Discussion. After background, describe your findings or observations under 'Results' section, followed by a section 'Materials and Methods' describing all methods used and describe materials used if any, for doing so, under a separate section. Please refer previous publications.
Hence I cannot recommend this manuscript in the present form.
Reviewer 2 Report
Unfortunately I feel the authors rushed to publish without having a clear plan for their manuscript. Overall, it needs a native English speaker to edit it for grammar and spelling as it is difficult to understand in most places. The tables are not cohesive; one is shown without horizontal lines and one has the lines. The figures are not easy to understand. They are overlapping one another, the numbered letters (A, B, C) are barely legible as they blend into the photos. Using red to highlight on a dark image with a green background renders it completely invisible to those with color blindness. All images must be rethought and redesigned. Also there is a final image with no identification for the multiple images within it and no reason given as to its placement. Above all, there is no Material and Methods section, nor a Results section. The figures and tables are all part of the introduction which is quite confusing. To round out the paper, the authors summarized their entire article with two paragraphs of discussion and one paragraph of conclusions. Of which, the conclusions gave no insight on what possibilities for remediation could exist or how they might further research the issue. The authors need to do some restructuring and further analysis before submission again.